# Fire Severity Causes Temporal Changes in Ground-Dwelling Arthropod Assemblages of Patagonian *Araucaria–Nothofagus* Forests

**Alfredo H. Zúñiga** [1,2,3,4,*], **Jaime R. Rau** [2], **Andrés Fierro** [5], **Pablo M. Vergara** [5], **Francisco Encina-Montoya** [6], **Andrés Fuentes-Ramírez** [7,8] **and Fabian M. Jaksic** [4]

1    Departamento de Ciencias Agronómicas y Recursos Naturales, Universidad de La Frontera, Temuco 4810296, Chile
2    Laboratorio de Ecología, Departamento de Ciencias Biológicas y Biodiversidad, Universidad de Los Lagos, Osorno 5310000, Chile
3    Programa de Doctorado en ciencias mención conservación y manejo de Recursos Naturales, Universidad de Los Lagos, Puerto Montt 5480000, Chile
4    Center of Applied Ecology and Sustainability (CAPES), Santiago 8331150, Chile
5    Departamento de Gestión Agraria, Facultad Tecnológica, Universidad de Santiago de Chile (USACH), Santiago 9170022, Chile
6    Núcleo de Estudios Ambientales, Departamento de Cs. Ambientales, Universidad Católica de Temuco, Temuco 4810296, Chile
7    Laboratorio de Biometría, Departamento de Ciencias Forestales, Facultad de Ciencias Agropecuarias y Medioambiente, Universidad de La Frontera, Temuco 4810296, Chile
8    Centro Nacional de Excelencia para la Industria de la Madera (CENAMAD), Pontificia Universidad Católica de Chile, Santiago 8331150, Chile
*    Correspondence: alfredo.zuniga@ufrontera.cl

**Abstract:** Fire is one of the main drivers of anthropogenic disturbances in temperate forest ecosystems worldwide, with multiple effects spread across ecological networks. Nevertheless, the biodiversity effects of fire are poorly known for species-diverse groups such as arthropods. In this research, we used a burn gradient generated two and three years after a large fire event to assess how different levels of fire severity affect arthropod diversity in the forest with the main forest forming long-living tree species *Araucaria araucana*, in southern Chile. The species richness and abundance of arthropods among trophic guilds was estimated annually in four fire-severity levels. We found that arthropods responded differently to fire severity levels, depending on their trophic guilds and years after fire (two and three years after fire). During the second year after fire, zoophages, phytophages, and parasitoids were more diverse in areas with high fire severity within the second year after fire, as compared to those in areas with low severity or unburned stands. In the third year after fire, a change in this trend was observed, where the abundance of all groups dropped significantly, with positive changes in the diversity in zoophages, phytophages, polyphages and saprophages, which is more noticeable in sites with high severity. These results indicate that annual variation in environmental conditions triggers bottom-up cascading effects for arthropods. Forests stands severely impacted by fires support highly fluctuating and possibly unstable arthropod assemblages. Hence, restoration efforts should be focused on recovering microhabitat conditions in these stands to allow the persistence of arthropods.

**Keywords:** *Araucaria araucana*; disturbance; epigeic entomofauna; disturbances; National Reserve; trophic guild

## 1. Introduction

The distribution of temperate of forests globally is determined in part by fire [1], which affects vegetation dynamics, soil properties and species composition of forests [2,3]. Still, wildfires have become increasingly frequent, severe, and extensive under the current climate change conditions, interacting synergically with land use change and human

activities [4]. Although the increased frequency and severity of forest wildfires threaten several forest-dwelling species [5], the sensitivity of each taxon to wildfire depends, among other things, on the ability to disperse, ecological flexibility to use fire-generated resources and reproductive performance in burned areas [6–8].

Some species may serve as ecological indicators of fire-affected areas [9,10]. Nonetheless, species in taxonomically diverse groups, such as arthropods, may differ markedly in their responses to wildfire [11,12]. Some arthropods respond positively to the increased resources available during the recovery phase of vegetation postfire, whereas others face an inhospitable habitat [13,14]. The heterogeneous species-specific response of arthropods to wildfires may substantially modify the trajectory of the assemblage in burned forests [15]. Hence, making distinctions among species that respond differently to fire regimes provides an operational framework for understanding changes in arthropod assemblages at broader spatio-temporal scales [16,17]. In forest ecosystems, arthropods play multiple ecological roles that are closely related to the community's trophic guild structure [18–21]. Some arthropod trophic guilds are sensitive to wildfire depending on their severity by affecting the dynamics of their food resources [22,23]. Predatory, mycophagous, and detritivorous arthropods usually are negatively affected by wildfire [24,25], whereas xylophages, phytophages, and parasitoids tend to be favored by wildfire [25,26]. Nevertheless, it is still unclear how different guilds are affected by wildfires [27], especially within relict, old-growth forests with high ecological, conservation and cultural values.

*Araucaria–Nothofagus* forests in Chilean Patagonia have been increasingly affected by severe wildfires in recent decades [28]. Although forest fires in this region are naturally infrequent and normally of low severity [29], the risk of larger and more devastating wildfires is being amplified by the effects of regional droughts [30]. The historical low frequency of large wildfires in Chile's forests has been put forth to explain the high sensitivity and low adaptive capacity of such vegetation to wildfire [31]. The increasing frequency, magnitude, and severity of wildfires could largely modify the natural forest dynamics and the entire ecosystem functioning [32], with cascading effects in food webs [4]. So far, few studies have reported the response to wildfire of arthropods in Patagonian forests. In Argentina, wildfires have been shown to cause a decrease in the ground species richness of beetles (Coleoptera) [24,33,34], but their effects on saproxylic [25] and ground beetles [24] differ. In Chile, arthropod assemblages exhibit important interannual variability [35], and wildfires in *Araucaria–Nothofagus* forests have been shown to cause a transient increase in deadwood and litter, with a concomitant postfire decrease in understory cover and tree density [8,36]. Thus, more advances in understanding the effects of wildfire on the arthropod assemblages of *Araucaria–Nothofagus* forests are needed to determine how their trophic guild structure responds to postfire changes in habitat conditions. We assessed the effects of a recent high-severity wildfire (occurred in 2015) on the trophic structure of ground-dwelling arthropod assemblages in a protected area, dominated by *Araucaria–Nothofagus* forests in southern Chile. Our study focused on stands with different fire severity levels, which were continuously monitored for two years after the wildfire (during 2017 and 2018). We hypothesized that high levels of wildfire severity in a protected area of southern Chile modify the trophic structure of arthropod assemblages, decreasing the abundance and diversity of their component guilds. In addition, we tested the hypothesis that the diversity of arthropods changes over time, as a consequence of the processes of forest recovery patterns from fire.

## 2. Materials and Methods

### 2.1. Study Site

This study was carried out in the National Reserve China Muerta (38°42′00′′ S–71°26′00′′ W; 1550 m a.s.l.), within the Andean range in southern Chile (Figure 1). The reserve encompasses 11,168 ha, with the domination of *Araucaria araucana* (monkey puzzle tree) and associated *Nothofagus pumilio* (ñirre) tree species [37]. The understory consists of the shrubs: *Chusquea culeu*, *Maytenus distichia* and *Gaultheria poepiggi;* the herbaceous

layer consists of *Osmorhiza chilensis*, *Viola magellanica* and *Adenocaulon chilense* [38]. Within our study area, *A. araucana* populations occur in pure (>90% canopy cover) or mixed with *N. pumilio* stands. The climate is warm with dry summers and snowy winters [39]. Soils were formed by recent volcanic ashes (Andisols and Histosols orders), and are well stratified, deep, and dark brown in color, with a coarse texture and displaying permeability throughout the profile [40].

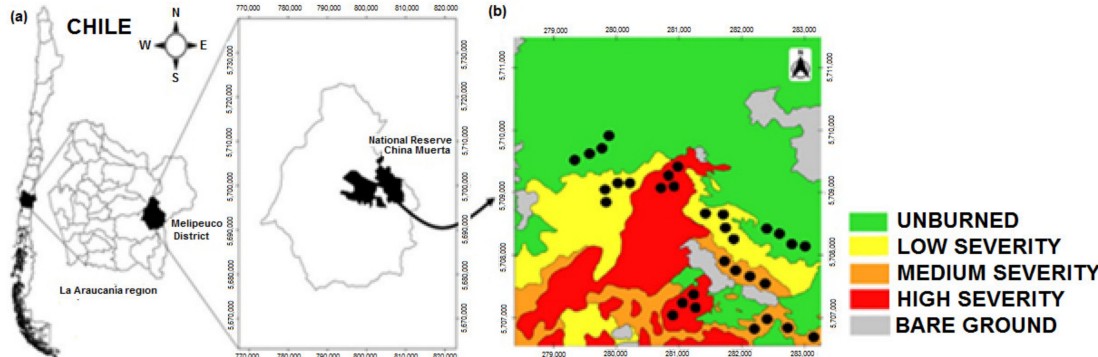

**Figure 1.** Map of the study site in the Andes ranges of southern Chile (**a**), including the burn severity conditions where arthropods were sampled (**b**). Black dots indicate pitfall traps set in triangular arrangement.

## 2.2. Sampling Design

During the austral summer and fall of 2015, the study area was affected by a large wildfire covering a total of 3700 ha, with the burn spreading in a mosaic pattern with different degrees of fire severity [41]. Based on aerial images and qualitative evaluation in the field [42], eight study plots were selected, with more than 300 m from each to reduce spatial dependence. Each of the two plots was assigned to one of four levels of wildfire damage: (1) unburned, forest without signs fire damage; (2) low severity, with superficial burning of the vegetation and without root damage; (3) medium severity, including partial crown damage and root damage; and (4) high severity, with total combustion of shrubs and high tree mortality (i.e., carbonization), and also with the presence of underground fires. The design followed a fire severity gradient previously used in other studies at the same site [36,42,43]. Plots had an average size of 2 ha.

## 2.3. Arthropod Data

The austral summers of 2017 and 2018 (hereafter years 1 and 2 of the study; two and three years postfire), 96 pitfall traps were installed in each study site. Each trap was 95 mm deep and consisted of a 50 mm-diameter plastic container, with a mixture of water, glycerin, and 95% ethanol, in a proportion of 10:1:1 [43]. In each severity condition (low, medium, high, and unburned), two linear sets of four trapping stations, each equipped with three traps arranged in triangular pattern, were placed in the field [44] (Figure 1). Trapping stations were >100 m apart from each other to reduce spatial autocorrelation. Traps were installed in January of 2017 and of 2018 and checked after two weeks. The individuals captured in each group of three traps were pooled and considered to be one statistical sampling unit for later analysis of abundance and diversity (see below). After checking traps were removed from the field, their contents were poured into jars with 70% alcohol, for the subsequent taxonomic identification of arthropods in the laboratory. Arthropod individuals were classified at the family level by consulting taxonomic keys, specific literature, and collections in Chile's Museo Nacional de Historia Natural in Santiago. Family-level identification yields unbiased community descriptors for diverse taxonomic groups such as macroinvertebrates [45]. Based on the trophic guild classification for each family, trapped individuals were assigned to one of six trophic groups: zoophages, phytophages, polyphages, parasitoids, saprophages, and mycophages [46,47].

*2.4. Data Analysis*

We determined the effect of fire severity on the taxonomic diversity of trophic guilds by comparing diversity metrics derived from curves of rarefaction and extrapolation [48–50]. From these curves, Hill's series of diversity indices were estimated for each fire severity treatment [51]. The following Hill numbers of different orders (q) were estimated using the iNext package in the R environment: q = 0, richness of families; q = 1, exponential of Shannon's exponential entropy index; and q = 2, multiplicative inverse of the Simpson index [51]. These indices are based on the proportion of the different components of the assembly, which allows obtaining an estimator of the degree of heterogeneity that they present [52]. Shannon's index is sensitive to the number of rare species, whereas Simpson's index is sensitive to the dominance of abundant species [51]. Completeness of sampling was evaluated as the sample coverage proposed by Chao and Jost [53]. The 95% Confidence Intervals (CI) for diversity indices were estimated from bootstrapping 1000 replicates and differences in diversity metrics were interpreted by comparing the mean of these distributions [54]. The abundance (i.e., the total number of individuals recorded in each trapping stations) of the six trophic groups was evaluated using generalized linear mixed models (GLMMs) [55], implemented with the lme4 package in R. Abundance data were assumed to have negative binomial distributions because evidence of overdispersion was detected in the data, using the AER R package. The effects of fire severity, sampling year and their interaction (Year × Severity) were evaluated with a log-link function. We included the interaction between year, replicate (forest stand), and fire severity as a random effect.

## 3. Results

A total of 37,340 individuals of three Arthropoda classes (Insecta, Entognatha, and Arachnida), from 90 families, were collected (30,641 individuals in 74 families in 2017; 6699 individuals in 65 families in 2018; 50 shared families present in both years). The diversity of arthropods varied greatly across fire severities, between trophic guilds, and over the two years of study (Table S1). Among zoophages, the families Cantharidae, Carabidae and Staphylinidae were the most prominent, while in the case of phytophages, it was the family Formicidae. For mycophages, Myceptophillidae was the family with most representation, and in the case of parasitoids, Phoridae was the dominant family. For polyphages, Chrysomelidae was the most abundant across family treatments, and for Saprophages, Collembolla (springtails) was the group with the greatest dominance in relation to other guilds. During 2017, we observed a positive response to fire intensity for: richness of zoophages, phytophages, and parasitoids, and diversity of phytophages and mycophages (Figure 2). Conversely, we observed a negative response to fire severity of zoophages, and saprophages, although for the former both indices decreased only in areas of high fire severity (Figure 2). During 2018, we observed a positive response for the diversity of phytophages and mycophages, while indexes of zoophages increased only in areas of high severity (Figure 2). Conversely, we found a decline in the richness of polyphages, as well as for the diversity of polyphages and saprophages (Figure 2). Additionally, parasitoids, polyphages, and saprophages declined in richness and diversity in at least one of the severity levels (Figure 2). Nevertheless, diversity indices of parasitoids, polyphages and saprophages exhibited an increase in either the low or medium fire severity conditions (Figure 2). When comparing 2017 and 2018, we observed a decrease in the diversity of mycophages and an increase in saprophages (Figure 2). In addition, diversity changes between years were more marked in the high-severity treatment, as compared with the control. The richness of zoophages, phytophages and polyphages as well as the diversity indices of parasitoids, polyphages and saprophages decreased in 2018 within areas of high fire severity (Figure 2). Conversely, the diversity of zoophages, phytophages, and mycophages increased in a later year in high-severity stands (Figure 2).

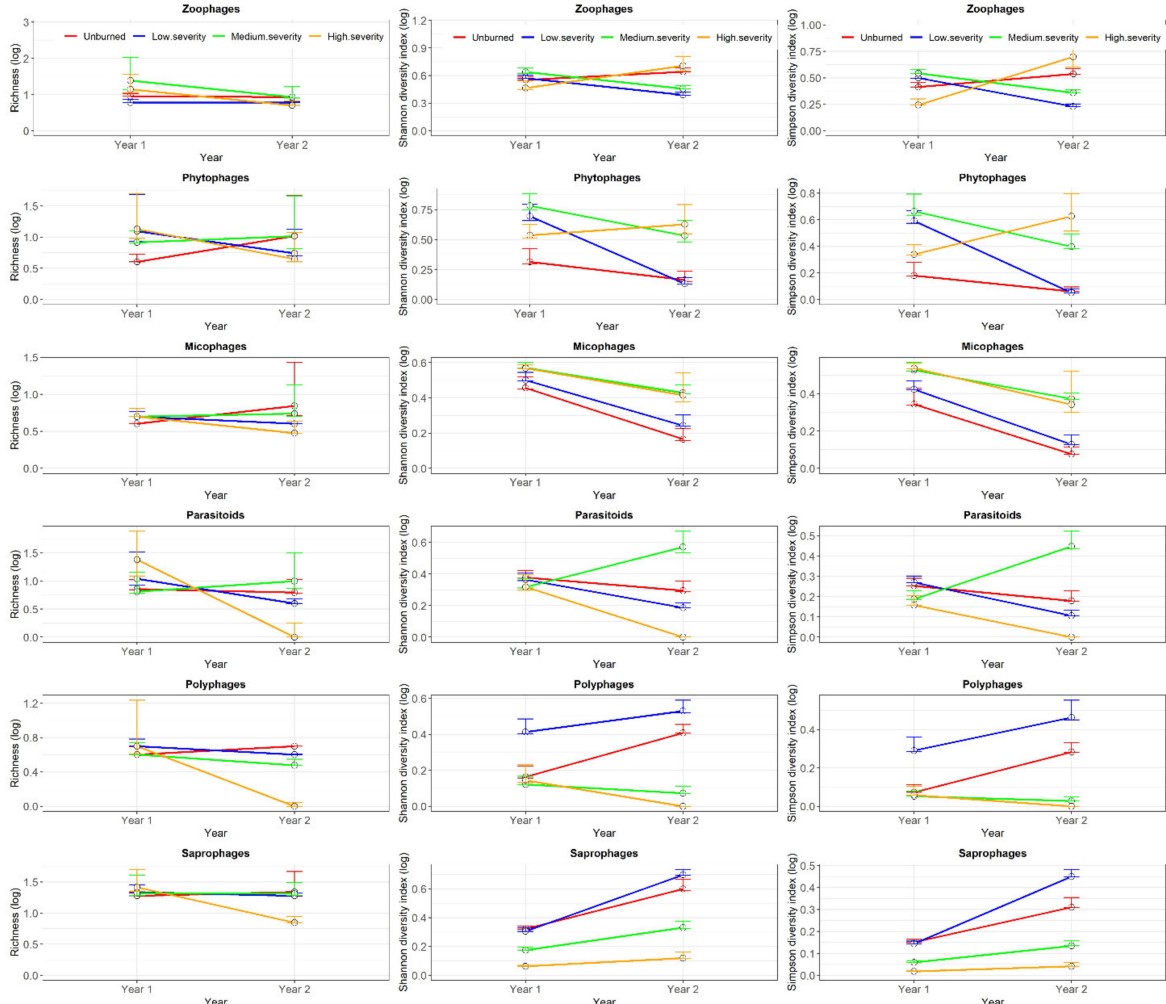

**Figure 2.** Diversity indices (richness, Shannon, and Simpson) of arthropods belonging to different guilds under different fire severity treatments collected during two sampling years (2017 and 2018).

The abundance of all trophic guilds of arthropods experienced marked changes between fire severity conditions after wildfire (Figure 3), and these changes were captured by the Bayesian model coefficients (Figure 4). Zoophages were the one guild whose abundance decreased in areas of high fire severity in comparison to the unburned stands (Figures 3 and 4). In contrast, saprophages and phytophages were the only guilds whose abundance increased in areas of high fire severity compared to unburned stands (Figures 3 and 4). Polyphages and mycophages increased in abundance at medium-fire-severity regimes (Figures 3 and 4). Nonetheless, the effects of fire on the abundance of these groups differed between years. Compared to 2017, in 2018 all trophic guilds tended to decrease in at least one of the severity conditions with reference to unburned stands (Figures 3 and 4). Negative interactions between fire and year were found in: (1) zoophages and saprophages (year × high severity) and (2) polyphages, mycophages, parasitoids, and phytophages (year × high severity and year × medium severity), all of which exhibited a pronounced decline in abundance during 2018 (Figures 3 and 4). In contrast, the positive interaction of year × low severity found for zoophages resulted from an increase in their abundance during 2018 in the low-severity stands (Figure 3).

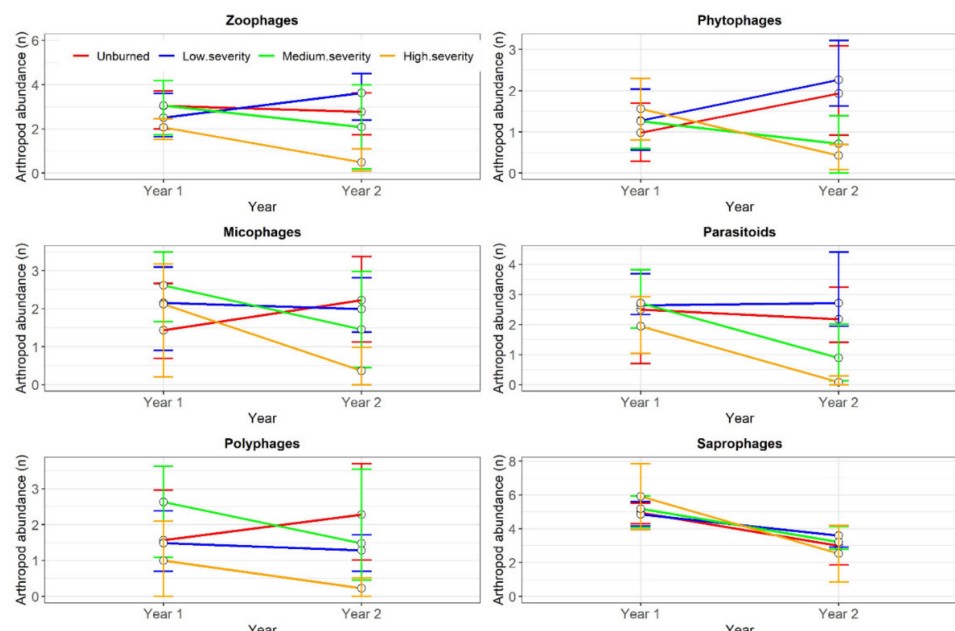

**Figure 3.** Abundance of arthropods belonging to different guilds under different fire severity treatments collected during two sampling years (2017 and 2018).

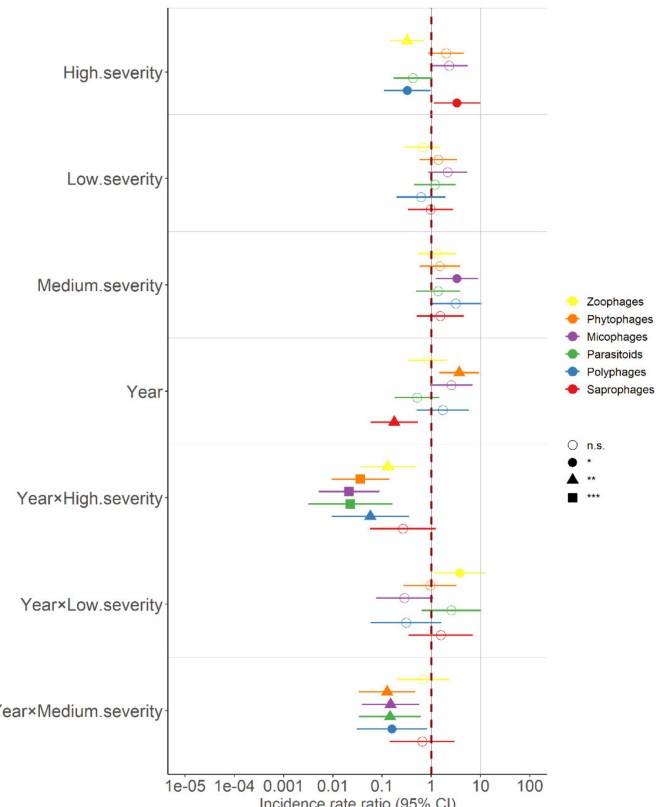

**Figure 4.** Incidence rate ratio (IRR) of GLMM fixed-effect coefficients for the effects of wildfire treatments (low severity, medium severity, and high severity), year, and the interaction of both factors on the abundance of arthropod guilds. IRR values represent the difference in the logs of expected abundance when the predictor variable changes in one unit. Significant effects are those where the intervals of coefficients do not intersect the red line (IRR = 1). Coefficients for the first year and unburned (control) treatment were considered as contrasts (assigned with the value 0) to test for differences among the levels of those factors.

## 4. Discussion

Fire severity influenced the forest arthropod assemblages examined, but its effect varied strongly among trophic guilds, in diversity metrics and over time. In terms of diversity, our research showed that only saprophages were negatively affected by wildfires over the two years sampled, while the abundance of zoophages, parasitoids, polyphages, mycophages, and phytophages decreased in burned areas over the two sampling years. Saprophages were numerically dominated by springtails (Collembola), which are known to avoid the effects fire or shelter in unburned sites or take refuge in deeper soil layers [56]. Thus, the loss of diversity was accompanied by the increased abundance of springtails in the areas of high severity.

The changes observed in the composition of the arthropod assemblages in each treatment could be explained in the first instance by the loss of their microhabitats through fire, which affects the specific food and shelter requirements of the species [57]. Consequently, many families of arthropods were found to be absent in the most severe sites, which shows the restrictions for their use. Second, the changes are attributable to the emergence of new groups of species, attracted by the new reorganization of resources generated by the disturbance, with which they are favored [58], a situation that is applicable in the case of fallen logs, which were found in greater numbers in high-severity sites [8]. Although the proportion of families that presented a positive relationship with severity was low, it is possible to assume a pattern of change in the composition of the assemblages, thus differing from the unburned condition. This situation would partially explain the positive relationship between the increase in family richness in some unions and the decrease in their respective abundances, which is consistent with that reported in coniferous forests in the Northern Hemisphere [26]. Thus, a decrease in dominance in each guild would result in a relaxation in the competitive interactions of the most abundant members, which would allow the occurrence of less abundant groups [59].

Zoophages were also negatively affected by wildfires, with a reduction in abundance rather than richness. Further, the diversity of zoophages in areas of high fire severity exhibited a contrasting pattern, with the lowest diversity found in the first postfire year and the highest in the second year. Zoophages are known to be sensitive to fire disturbances [60]; hence, their numerical reduction in the stands severely impacted by fire could result from prey limitation, microhabitat loss, or dispersal limitation [61]. Zoophages such as Staphylinidae, Cantharidae, and Carabidae were the most abundant in this guild, whose larvae inhabit mostly in ground, litter, or woody debris, microhabitats usually altered or destroyed by wildfire. Forest-dwelling Carabidae such as *Ceroglossus* (Solier) and *Creobius* (Guerin-Menevielle) are wingless with reduced dispersal ability and preyed on by birds and lizards [62,63], which may result in low colonization rates of the high fire severity stands. Moreover, zoophages likely faced a reduction in prey availability during the second postfire year, judging from the decreased abundance of most other arthropod guilds. The particularly high diversity of zoophages during 2018 may have resulted from an important numerical decline of Cantharidae beetles in response to the low availability of prey. In the case of parasitoids, all diversity indices decreased markedly in the high severity stands during the second postfire year, a pattern also followed in their abundance.

Hymenopterous parasitoids are vulnerable to decreased prey availability in the burned stands, as observed in saprophagous beetles, which decreased in abundance the second postfire year. Our results contrast with those of previous studies showing an increased diversity of parasitoids after wildfire, arguably resulting from creation of new microhabitats [60]. In our case, the new substrates (e.g., forest debris) available after wildfire possibly did not host diverse or abundant saprophages (e.g., springtails). Polyphages, mostly ants, responded to wildfire similarly to parasitoids, likely being influenced by the high seed production of annual plants [64].

The abundance, richness, and diversity of mycophages exhibited contrasting responses to wildfire over the two years of study. Indeed, during the second year mycophages increased their diversity while reducing their abundance, likely resulting from the lowered

numbers of the most abundant species (Latridiidae and Mycetophilidae). Unsuitable climate conditions during the first year, such as high summer temperatures, possibly contributed to the low production of fruit bodies. Indeed, the year effect was notorious in high-severity stands where fungal substrates are more exposed to sunny and extreme heat conditions in summer. Previous studies indicate that fire effects on fungi are mediated by several environmental factors. For instance, some fungal species respond positively to the postfire susceptibility of trees to being attacked by root and canker pathogens [65]. Conversely, saprotrophic fungi living in woody debris and leaf litter are severely affected by wildfires [66]. Similarly, annual variation in the biomass of annual plants may have resulted in low richness and abundance of phytophages found in the high-fire-severity stands. The abundant Cicadellidae and Curculionidae experienced decreased abundance in those stands highly affected by wildfire.

Our results point out that annual variation in environmental conditions may trigger bottom-up cascading effects on assemblages of arthropods. We propose that stands severely impacted by wildfire support highly fluctuating and possibly unstable arthropod assemblages resulting from temporally variable metacommunity parameters (e.g., colonization and extinction rates; [67,68]). However, our results are based on a single wildfire event; hence, the inference of fire effects on wildfire and the possible consequences of these effects on the restoration of Araucaria–Nothofagus forests in northern Patagonia should be further researched. For instance, restoration efforts should focus on improving microhabitat conditions within high-fire-severity stands. Increasing dead wood availability should favor the presence of saprophagous at a local scale, with indirect effects on the other guilds, thus speeding up the processes of forest succession.

**Supplementary Materials:** The following are available online at https://www.mdpi.com/article/10.3390/fire5050168/s1. Table S1. Total abundance (+ standard deviation among traps) of the families recorded in the study area through the treatments and sampling periods (year 1 and 2). The trophic guild to which they belong is also indicated. Mc: Micophages; Zp: Zoophages; Pl: Phytphages; Pt: Polyphages; Ps: Parasites; Sp: Saprophages.

**Author Contributions:** Conceptualization: A.H.Z., J.R.R., F.M.J. Data curation: A.H.Z., A.F. Formal analysis: P.M.V., F.E.-M. Investigation: A.H.Z. Methodology: A.H.Z., J.R.R., F.M.J., A.F. Funding acquisition: A.F.-R., F.M.J. Supervision: J.R.R., F.M.J. Writing-original draft: A.H.Z., P.M.V., A.F. Writing-review and editing: J.R.R., F.M.J., A.F.-R. All authors have read and agreed to the published version of the manuscript.

**Funding:** This research was partially funded by FONDECYT 11150487 and DIUFRO DI20-0066. A.F.R. thanks the support received from ANID BASAL FB210015 (CENAMAD), ANID SCIA-Anillo ACT210052 and Universidad de La Frontera DI22-1003. PMV tranks the support received from Proyecto DICYT, 092275WE_Ayudante.

**Data Availability Statement:** Data are available by request to the corresponding author.

**Acknowledgments:** We are grateful for Park rangers from the National Reserve China Muerta that helped us with the fieldwork. We also thank reviewers that made a number of comments and suggestions which greatly improved the manuscript.

**Conflicts of Interest:** There are no conflict of interest.

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
