# Peer review of "Fire Severity Causes Temporal Changes in Ground-Dwelling Arthropod Assemblages of Patagonian Araucaria–Nothofagus Forests"

_fire, doi:10.3390/fire5050168_

Round 1

Reviewer 1 Report (Previous Reviewer 2)

Dear Authors:  Thank you for your efforts at revisions. I've attached a document with my original comments, your responses, and my responses to your responses.

I appreciate the work you've done.

Author Response

Reviewer 2 Report (Previous Reviewer 3)

After reading the submitted paper carefully, I conclude that most of the comments included in the review were taken into account, and the quality of the paper was substantially improved. Nevertheless, I have found some minor errors (please see below) that should be considered before the manuscript is accepted for printing. I also suggest to the authors for the future that the changes introduced in the text should be marked in red, which greatly facilitates the review of the article.

Line 99: please remove „the study”

Lines 101-102: remove „forests”; latin names of plants should be written in italics

Line 103: instead “is dominated” use “consists of”

Lines 103-106: Latin names of plants should be written in italics

Line 115: please add the area (ha) of study plots

Line 145: : still missing information abou trapping period. For how many months have insects been caught?

Line 155: zoophages

Line 191: richness

Lines 209-221: different font type and size from the rest of the text

Line 215: remove one space before "Compared"

Line 217: zoophages

Line 218: polyphages

Line 220: change order positive interaction of year × low severity

Line 365: remove „According to us”

Lines 370-372: Still this part should be rewritten. You can’t say „thanks” or „acknowledge” in relation to the grant no. 

Author Response

Reviewer 3 Report (Previous Reviewer 1)

The manuscript has been significantly improved and now warrants publication in fire

Author Response

Thank you for your comments.

This manuscript is a resubmission of an earlier submission. The following is a list of the peer review reports and author responses from that submission.

Round 1

Reviewer 1 Report

To continue further studies of forest ecosystems' fire dynamics in Chile, due to getting such important and interesting for an international scientific community scientific results.

Author Response

No commentaries for further replies.

Reviewer 2 Report

Dear Authors:  I appreciate the work that has gone into this paper.  I have attached some comments that I think will improve the paper.

Author Response

Comments on fire-1803158

REVIEWER 2

General:
This paper is generally well-written and will, with appropriate revisions, make a contribution to the literature. Specific comments follow:
Specific:
Line 48ff: Generally, the authors fail to distinguish between “fire” and “wildfire,” and they use these terms interchangeably—this is problematic. Although prescribed fires are anthropogenic (they are, by definition, fires that are conducted and controlled under prescription), wildfires are not anthropogenic (see, e.g., among many other possible citations, Hiers et al. 2020. Prescribed fire science: the case for a refined research agenda. Fire Ecology 16:11 https://doi.org/10.1186/s42408-020-0070-8). Thus, if the first sentence is going to cite Bond et al., I would (strongly) suggest that the paper begin with:
“The distribution of temperate forests globally is determined in part by fire [1], which affects vegetation dynamics, … [2,3].”
This is consistent with the abstract, too.
The second sentence can, if the authors wish, segue into the topic of wildfires—which are not anthropogenic, with something like:
“However, wildfires have become increasingly frequent…”
Response: We used the term “wildfire” throughout the manuscript when used as a substantive. We also modified the first part of the Introduction as suggested.

Line 52: maintain consistent citation format—i.e., either use a format with [citation number] or a format with “author and year” format.
Response: We improved citation format

Line 53: change “and severity of forest fires” to “and severity of forest wildfires”
Response: Changed as suggested

 Line 57: change “fire affected areas” to “fire-affected areas”
Response: Changed as suggested

Line 60, 69: change “while” to “whereas”
Response: Changed as suggested

Line 94ff: Can you make your hypothesis more specific? To say that arthropod abundance and diversity “change” with level of fire severity is astonishingly vague—can you (a priori) hypothesize an increase? A decrease? What does the literature suggest?
Response: We have reformulated the hypothesis to be more specific in describing the fire effect

Section 2.2:
Line 114: was the fire a wildfire or a prescribed fire? If the latter, under what conditions (air temp, RH, windspeed, was the fire conducted?)
Response: We explain that different levels of “wildfire” damage were studied

Line 116ff: Clearly, there was just one fire and so, strictly speaking, the 8 plots (or the sets of 2 plots, depending on severity) cannot be considered true replications; and thus the study, generally, suffers from pseudo-replication (sensu Hurlbert; see below). This cannot be avoided with this kind of research, but it needs to be acknowledged and clearly pointed out by the authors so that readers are aware.
Response: We appreciate the comment about the need to avoid pseudoreplication in field experiments like this.  However, we think our sampling design and statistical analysis were appropriate to prevent temporal and spatial pseudoreplication. Our reasoning supporting this interpretation is based on Hurlbert et al. (1984), who holds that pseudoreplication in mensurative experiments is "often a consequence of the actual physical space over which samples are taken or measurements made being smaller or more restricted than the inference space implicit in the hypothesis being tested." First, we disagree with the comment that treatment replicates "cannot be considered true replications". Treatment replicates were different forest plots located more than 300 m from each, each being considered as hosting a particular community of arthropods. We based this interpretation on the low dispersal capacities known for some of the arthropods collected in these stands. Indeed, within each sampling stand, trapping stations (groups of three traps) were >100 m apart, and data from each trapping station were pooled to reduce spatial dependence. Second, our hypothesis now focuses on the fire effects "in a protected area of southern Chile". Thus, the results of our study are local, and the generalization of them should consider other studies in the same ecosystems. The latter was discussed in the last paragraph of the Discussion section.

Section 2.4:
1. RE: lines 166 – 169 and use of overlapping confidence intervals to determine significance:
a) This is widely practiced but it is inappropriate for lots of reasons, not the least of which is the following: if all one had to do to judge whether (say) 2 means were significantly different was to compute confidence intervals about each mean and check to see if they overall, then why was a t-test invented? And for 3 or more means, why was an F test invented?
b) That this practice is inappropriate has been well-documented in the literature (e.g., Wester, D.B. 2018, Comparing treatment means: overlapping standard errors, overlapping confidence intervals, and tests of hypothesis. Biometrics & Biostatistics International J., 2018;7(1): 00192. DOI: 10.15406/bbij.2018.07.00192 and the many, many citations contained therein; Krzywinski and Altman, 2014, Visualizing samples with boxplots, Nature Methods, VOL.11 NO.2 FEBRUARY 2014; Belia, Fidler, Williams and Cumming, 2005, Researchers misunderstand confidence intervals and standard error bars, Psych. Methods, 10:389-396; Payton, Miller and Raun, Testing statistical hypotheses using standard error bars and confidence intervals, Communications in Soil Science and Plant Analysis, 31:5-6, 547-551,DOI: 10.1080/00103620009370458). There are dozens of other citations available to support this point.
c) Many commonly-used diversity indices (including what is used in this paper) do not fulfill normality assumptions underlying tests of significance in normal-based models (e.g., Fritsch, K.S. and Hsu, J.C. 1999. Multiple comparison of entropies with application to dinosaur biodiversity. Biometrics 55:1300-1305; Rogers, J.A. and Hsu, J.C. 2001. Multiple comparisons of biodiversity. Biometrical J. 43:617-625. Lande, R. 1996. Statistics and partitioning of species diversity, and similarity among multiple communities. Oikos 76:5-13. Bowman, K. O., K. Hutcheson, E. P. Odum, and L. R. Shenton. 1971. “Comments on the Distribution of Indices of Diversity.” In Statistical Ecology, edited by E. C. Patil, E. C. Pielou, and W. E. Waters, 3:315–66, University Park, PA: Pennsylvania State University Press). Thus, the analysis employed by the authors using Bayesian inference and normal non-informative distributions as priors is not appropriate. Also, it is unclear why a generalized linear mixed model was used, and what link function was employed and why. The authors mention log-transformation— log-transforming abundance and richness data is appropriate, but this involves transformation of the response variable rather than mean of the response variable through an appropriate link function, and so much of this is unclear in the mss and possibly inappropriate.
d) More details are needed about how the repeated measures aspect of the analysis was handled. That is, sampling over time often introduces temporal non-independence; in linear mixed models this can be handled on the R-side of the model with an appropriately-selected variance-covariance structure; none of this is presented in the mss.
Response: We thank reviewer# 2 for this comment about confidence intervals. We agree diversity indices usually do not fulfill normality assumptions and treatment comparisons should be carried out by using statistical tests. However, the following points should be clarified. First, we used rarefaction/extrapolation curves of diversity that consider a resampling procedure for confidence intervals. We used data from all sampling units to characterize the diversity of each treatment into a single diversity estimate for each Hill number. Thus, the diversity measurement of each treatment is not averaged from sampling points but rather Hill numbers of species richness and diversity are extrapolated for the complete assemblage. Unfortunately, no statistical test is available in the iNEXT R package to compare diversity between communities, but such comparisons are made based on bootstrapping confidence intervals, as used in other studies. Second, we used neither Bayesian inference nor generalized linear mixed models to analyze the diversity of arthropods, hence the comment about using repeated measures in models is not applicable. We think some confusion there exists in the manuscript when explaining the methodologies used to compare diversity and abundance data. Thus, we rewrote this section to clarify the analyses used.

e) My suggestions are as follows:
i) Use a general linear mixed model with the following clearly defined. 
Response: We used a general linear mixed model (GLMM) to analyze abundance data. GLMM was implemented with a log link function and negative exponential distribution. Models now include a nested-error structure, where temporal (year) random effect was nested within the plot (forest stand).

ii) There was, actually, only one wildfire in this project, and so the 2 severely- burned areas are not true replications but rather pseudo-replications of the “severely-burned” condition; similar comments apply to the 2 non-burned areas, etc. That is, the authors should acknowledge that there is only one wildfire in this research and that 2 plots in each severity level are actually pseudo- replications (see S.H. Hurlbert’s classic monograph, Pseudoreplication and the design of ecological field experiments. Ecol. Monog. 1984, 54:187-211).
Response: As explained above, we disagree with the comment that treatments were not replicated because sampling stands were spatially independent, and our hypothesis raises the effects of a large wildfire in a protected area in southern Chile.

Regards Results: 
1. Based on the above comments, it is difficult to evaluate the results as presented. However, some general comments follow:
2. The graphs of the results have severity levels along the horizontal axis with solid lines connecting a given response variable from one severity level to another; and with 2 lines (one for each year of data). This implies that severity level is a continuous variable—a line connects one to another—when in fact it is discrete or categorical. 
a. I (strongly) suggest putting year on the horizontal axis, because (in some sense) going from year 1 to year 2 is a continuous change (albeit sampled at just 2 points). And then have 4 such lines, one from each severity level.
b. As shown, each solid line in the graph implies a continuous change from one severity level to another—but severity levels, as defined, are not continuous but categorical.
c. If the authors want to keep severity level on the x axis, then they should use paired bars, with members of each pair being yr 1 and yr 2).
Response: We thank reviewer#2 for this valuable recommendation. We modified the figures comparing diversity and richness (Figs. 2 and 3) between sampling years (horizontal axis), with each fire severity treatment now being distinguished with a different line color.  

3. There is quite a bit about Figure 4 that needs attention:
a. R2 values in generalized linear mixed models are not clearly defined, and their objective definition is not well accepted (e.g., Piepho, 2018, A coefficient of determination (R2) for generalized linear mixed models, Biometrical Journal, DOI: 10.1002/bimj.201800270;
Jaeger, Edwards, Das and Sen , An R2 statistic for fixed effects in the generalized linear mixed model. Journal of Applied Statistics, 44:6,1086-1105, DOI: 10.1080/02664763.2016.1193725; Stoffel, Nakagawa and Schielzeth, 2021, partR2: partitioning R2 in generalized linear mixed models, PerrJ, e11414 http://doi.org/10.7717/peerj.11414). A citation to the appropriate primary literature rather some an R package is needed.
Response: We analyzed the abundance of arthropods with GLMMs rather using Bayesian models. We did not interpret model goodness of fit based on R2 due to the reasons explained above by reviewer#2.

b. It will be very difficult for the average reader to interpret the coefficients and their CIs in Fig. 4. If you are going to present a figure like this, then it will be very important to explain how to interpret it—e.g., what does the coefficient for year × medium severity mean; and this will require explaining how dummy variables are defined so that the reader will understand why (for example) there are no coefficients for the unburned treatment; similarly, there is one coefficient for year—is this for year 1 or for year 2? Of course, it depends on the package used, so that, if year has only one coefficient, what exactly does this coefficient mean? Well, it represents a difference…but the average reader may not realize this, may not know what to make of this.
Response: In the legend of Fig. 4, we explained the meaning of the response variable (incidence rate ratio of coefficients) and the levels of factors included in GLMM (year and fire treatments). We also explained the coefficients for the first year and unburned (control) treatment were considered as contrasts testing for differences among the levels of those factors.

c. For what Fig. 4 is intended to convey, it will be much for informative to simply present abundance means (again, if they are back-transformed from a log(Y+1) transformation, they are really estimates of medians) together with P values for comparisons among severity levels, comparison among the 2 years, and if there’s an interaction involved, the appropriate simple main effects involved and the associated P values. In short, Fig. 4 needs to be replaced with something that the average reader will understand.
Response: As explained in the response to the above comment (b), now Fig 4 was improved to show the effects of year, fire treatments, and their interactions. We think this figure is needed due the interaction between year and fire severity is not easily appreciable from the abundance plots (Fig. 3)  

Regarding Discussion: As with the results, it is difficult to assess the Discussion section—that is, both Results and Discussion will need to be revised and re-evaluated when the data have been re-analyzed.

Response: An issue about environmental changes linked to sensitivity in arthropods was added.

Reviewer 3 Report

The paper “Fire severity causes temporal changes in ground-dwelling arthropod assemblages of Patagonian Araucaria-Nothofagus forests” has potentially interesting data on the effect of fire on the quantitative-qualitative structure of arthropod assemblages after two and three years from the event. Definitely the topic is important, due to the role of ground-dwelling arthropods in forests ecosystems and the lack of knowledge about the ecological aspect of the fires impact and their intensity on pedofauna communities. However, I found some weak points, which should be improved before acceptation manuscript for the publication.

The construction of the manuscript was well designed. The hypotheses and goals of the study, included in the Introduction section, correspond with each other and presented results. Nevertheless, due to the analysis of arthropod communities succession, also in relation to the time that has elapsed since the fire it is necessary to formulate a hypothesis based on the available knowledge. Moreover, the authors erroneously write about the effects of the fire in the perspective of one and two years, where it took place two and three years after this event. Perhaps the month in which the fire occurred may indicate the correctness of the years used, but it was not included in the methodological part. The methodology has been aptly chosen for the research objectives, although it requires in some areas of detail, like explanation what ecological indicators were used to characterize the arthropod assemblages and how they were calculated. Interpretation of the results is correct, although it should be expanded and described in more detail in the context of the identified families. This was included in the discussion, but omitted in the results part of the manuscript. In the discussion, although properly conducted, it lacks detailed practical conclusions from the research carried out. The lack of interpretation of the results based on differences in the environmental conditions of the study sites (as a consequence of different fire severity level), such as the presence of dead wood, may also be unsatisfactory. Taking above, I suggest the Editor to major revision of the manuscript and its re-reviewing after making appropriate corrections. All detailed comments and corrections are listed below.

Detailed comments:

Line 32: Rewrite the sentence: “...affect arthropod diversity in the forest with the main forest forming long-living tree species Araucaria araucana, in southern Chile.

Line 33: Delete “during 2015” as the effect of different degrees of forest severity was studied after two and three years from the event, not in 2015. Instead, you can add “two and three years after the event”.

Lines 33-35: Remove sentences” “The ecosystem...... Using pitfall traps.....treatments”.

Line 36: severity levels, depending

Line 38: if you mean 2017, this is two years after fire, not one year.

Lines: 38-39: When comparing areas did you mean in the same year? It is not clear in relation to areas with low fire severity and unburned areas. Please explain it.

Line 42: Remove “also”. When you mentioned about “improving” what do you mean? How? Please explain it.

Lines 45-46: Change keywords and order: China Muerta National Reserve, Araucaria araucana, fire, severity level, epigeic entomofauna, trophic guilds

Line 53: “wildlife species” is not precise. Did you mean insects, pedofauna?

Line 81: Instead „Still” I suggest to use „So far”

Line 82: after “beetles” add “(Coleoptera)”

Line 83: ground species

Line 91: “…in a protected area, dominated by…”

Line 96: I suggest adding the second hypothesis relating to the time impact on the arthropods occurrence. Probably the authors assumed the existence of differences. On what basis (literature review).

Lines 100-101: The study was carried out in the National…..1550 m a.s.l.), within the Andes range in southern Chile (Fig. 1).

Lines 102-103: The reserve…..11,168 ha, with the domination of Araucaria araucana (monkey puzzle tree) and associated Nothofagus pumilio (ňirre) tree species.

Line 103: (CONAF 2014) - ??? Not clear.

Line 104: the shrubs:

Line 105: instead “is dominated” use “consists of”

Line 106: Remove “can”

Line 107: pure (>90% canopy cover) or mixed with N. pumilio stands.

Line 114: When you mention about “mosaic pattern”, can you ensure that the study plots cannot be influenced by each other? It is difficult to deduce from Figure 1.

Line 116: Instead of term “forest stands” use “study plots” or “study sites”; “ …to conduct the arthropod specimens sampling.

Line 117: study plots or study sites were assigned

Line 122: Change order [36, 42, 43]

Line 139: In the austral summers; Why January to March? In the line 147 there is an information about installing traps in January. It is inconsistent. In addition, if the analysis was carried out in 2017 and 2018, and the fire took place in 2015, i.e. the effect was tested 2 and 3 years after the event, not one year and two years as stated.

Line 140: the number of traps installed at each study site should be added.

Line 140: “…traps were installed in each study site. Each trap was a 95 mm deep and consisting of 50 mm…”.

Lines 143-144: “…trapping stations, each equipped……in triangular pattern”

Line 147-148: There is a missing information about the period in which arthropods were collected.

Lines 148-149: Moved the sentence “Trapping stations were > 100 m apart….” in the line 144

Line 156: „….gropus: zoophages, ……”

Line 159: There is no information about the indicators used to describe the arthropod assemblages. The method of calculating the indicators: Shanon, Simpson, family diversity, and abundance should be explained.

Lines 181-220 This part should be expanded. The most frequently registered families and their dependencies on the degree of fire severity and timing of study should be described.

Lines 181-183: „…and Arachnida), 90 families…..were collected.

Lines 199, 200, 205: diversity, richness, abundace indices should be explained in the methodoloy section. See comment sbove.

Lines 262-263: Rewrite the sentence: „… but its effect varied strongly in diversity metrics among trophic guilds and over time.

Lines 267, 277-280: springtails (Collembola), Staphylinidae, Cantharidae, and Carabidae (Ceroglossus (Solier) and Creobius (Guerin-Menevielle) have never beed mentioned before. Shoul be described in result chapter (see my comment above).

Line 295: mostly ants

Kines 317-319: Practical conclusions resulting from the results obtained in the field of recommendations for the management of post-fire areas require expansion.

Lines 322-324: This part should be rewritten. You can’t say „thanks” or „acknowledge” in relation to the grant no.  

Author Response

Nevertheless, due to the analysis of arthropod communities succession, also in relation to the

time that has elapsed since the fire it is necessary to formulate a hypothesis based on the

available knowledge.

Response: Done

Moreover, the authors erroneously write about the effects of the fire in the perspective of one

and two years, where it took place two and three years after this event. Perhaps the month in

which the fire occurred may indicate the correctness of the years used, but it was not included

in the methodological part.

Response: A new hypothesis was stated as recommended

The methodology has been aptly chosen for the research objectives, although it requires in

some areas of detail, like explanation what ecological indicators were used to characterize the

arthropod assemblages and how they were calculated.

Response: We explained in detail the diversity indices used

Interpretation of the results is correct, although it should be expanded and described in more

detail in the context of the identified families. This was included in the discussion, but omitted

in the results part of the manuscript.

Response: We analyzed the trophic guild structure without making a taxonomic distinction.

However, we refer to the most important families at the beginning of the Result section

In the discussion, although properly conducted, it lacks detailed practical conclusions from the

research carried out.

Response: We included concluding management statements at the end of the Discussion, as

recommended

The lack of interpretation of the results based on differences in the environmental conditions

of the study sites (as a consequence of different fire severity level), such as the presence of

dead wood, may also be unsatisfactory.

Response: We delve into environmental conditions of wildfire treatments, focusing on dead

wood amounts

Line 32: Rewrite the sentence: “...affect arthropod diversity in the forest with the main forest

forming long-living tree species Araucaria araucana, in southern Chile.

Response: Done.

Line 33: Delete “during 2015” as the effect of different degrees of forest severity was studied

after two and three years from the event, not in 2015. Instead, you can add “two and three

years after the event”.

Response: DDate of event of fire was deleted and added, as suggested.

Lines 33-35: Remove sentences” “The ecosystem...... Using pitfall traps.....treatments”.

Response: “The ecosystem”, and “Using pitfall traps” were deleted. We retained

“Treatments” due to it is an necessary term describing the levels of fire severity.were deleted.

We think that “Treatments” is informative regarding levels of fire severity, where “control” is

also added.

Line 36: severity levels, depending

Response: Done, but ‘control’ was also added.

Line 38: if you mean 2017, this is two years after fire, not one year.

Response: Sentence relative to year was deletedWe clarified that comparisons were made

between two and three years after fire..

Lines: 38-39: When comparing areas did you mean in the same year? It is not clear in relation

to areas with low fire severity and unburned areas. Please explain it.

Response: A new sentence regarding changes in the second year of study was added. We

improved these sentences by pointing out those differences were tested within the same

years

Line 42: Remove “also”. When you mentioned about “improving” what do you mean? How?

Please explain it.

Response: Done.

Lines 45-46: Change keywords and order: China Muerta National Reserve, Araucaria

araucana, fire, severity level, epigeic entomofauna, trophic guilds

Response: Done.

Line 53: “wildlife species” is not precise. Did you mean insects, pedofauna?

Response: word.We replaced it by “forest-dwelling species”

Line 81: Instead „Still” I suggest to use „So far”

Response: Done.

Line 82: after “beetles” add “(Coleoptera)”

Response: Done.

Line 83: ground species

Response: Done.

Line 91: “…in a protected area, dominated by…”

Response: Done.

Line 96: I suggest adding the second hypothesis relating to the time impact on the arthropods

occurrence. Probably the authors assumed the existence of differences. On what basis

(literature review).

Response: A new hypothesis raising temporal changes in the diversity of arthropods , was

added.

Lines 100-101: The study was carried out in the National…..1550 m a.s.l.), within the Andes

range in southern Chile (Fig. 1).

Response: Done.

Lines 102-103: The reserve…..11,168 ha, with the domination of Araucaria araucana

(monkey puzzle tree) and associated Nothofagus pumilio (ňirre) tree species.

Response: Done.

Line 103: (CONAF 2014) - ??? Not clear.

Response: The This confusing term was deleted.

Line 104: the shrubs:

Response: Done.

Line 105: instead “is dominated” use “consists of”

Response: Done.

Line 106: Remove “can”

Response: Done.

Line 107: pure (>90% canopy cover) or mixed with N. pumilio stands.

Response: Done.

Line 114: When you mention about “mosaic pattern”, can you ensure that the study plots

cannot be influenced by each other? It is difficult to deduce from Figure 1.

Response: We clarify plots were more than 300 m from each to reduce spatial dependence.

Line 116: Instead of term “forest stands” use “study plots” or “study sites”; “ …to conduct the

arthropod specimens sampling.

Response: Done.

Line 117: study plots or study sites were assigned

Response: We used the term “study plots”.

Line 122: Change order [36, 42, 43]

Response: Done.

Line 139: In the austral summers; Why January to March? In the line 147 there is an

information about installing traps in January. It is inconsistent. In addition, if the analysis was

carried out in 2017 and 2018, and the fire took place in 2015, i.e. the effect was tested 2 and

3 years after the event, not one year and two years as stated.

Response: specified.We improved this sentence as suggested

Line 140: the number of traps installed at each study site should be added.

Response: The number of traps installed was now added.

Line 140: “…traps were installed in each study site. Each trap was a 95 mm deep and

consisting of 50 mm…”.

Response: Done.

Lines 143-144: “…trapping stations, each equipped……in triangular pattern”

Response: Done.

Line 147-148: There is a missing information about the period in which arthropods were

collected.

Response: We included this information in the first sentence of this paragraph

Lines 148-149: Moved the sentence “Trapping stations were > 100 m apart….” in the line 144

Response: Done

Line 156: „….gropus: zoophages, ……”

Response: Done.

Line 159: There is no information about the indicators used to describe the arthropod

assemblages. The method of calculating the indicators: Shanon, Simpson, family diversity,

and abundance should be explained.

Response: We mentioned theinterpretation of those diversity indices, but the

method of calculating the indicators is mentioned.

Lines 181-220 This part should be expanded. The most frequently registered families and

their dependencies on the degree of fire severity and timing of study should be described.

Response: This part was expanded as recommended.

Lines 181-183: „…and Arachnida), 90 families…..were collected.

Response: Done.

Lines 199, 200, 205: diversity, richness, abundace indices should be explained in the

methodoloy section. See comment sbove.

Response: The explanation of these indices was improved, as

mentioned in the above comment of reviewer#3

Lines 262-263: Rewrite the sentence: „… but its effect varied strongly in diversity metrics

among trophic guilds and over time.

Response: Done.

Lines 267, 277-280: springtails (Collembola), Staphylinidae, Cantharidae, and Carabidae

(Ceroglossus (Solier) and Creobius (Guerin-Menevielle) have never beed mentioned before.

Shoul be described in result chapter (see my comment above).

Response: Done.

Line 295: mostly ants

Response: Done.

Lines 317-319: Practical conclusions resulting from the results obtained in the field of

recommendations for the management of post-fire areas require expansion.

Response: Done.We included practical management conclusions at the end of the

Discussion section.

Lines 322-324: This part should be rewritten. You can’t say „thanks” or „acknowledge” in

relation to the grant no.

Response: We are mandated to acknowledge funding this way by the

respective agencies mentioned.